# Assessment of Genetic Stability in Human Induced Pluripotent Stem Cell-Derived Cardiomyocytes by Using Droplet Digital PCR

**DOI:** 10.3390/ijms25021101

**Published:** 2024-01-16

**Authors:** Ji Won Park, Su Ji Bae, Jun Ho Yun, Sunhee Kim, Misun Park

**Affiliations:** Advanced Bioconvergence Product Research Division, National Institute of Food and Drug Safety Evaluation, Ministry of Food and Drug Safety, Cheongju-si 28159, Republic of Korea; parkgeeone@korea.kr (J.W.P.); sujibaebae@korea.kr (S.J.B.); junho0970@korea.kr (J.H.Y.); skim427@korea.kr (S.K.)

**Keywords:** genetic stability, human iPSC-CM, exon sequencing, targeted panel sequencing, tumorigenicity, droplet digital PCR, mutation detection sensitivity, cell therapy, quality control

## Abstract

Unintended genetic modifications that occur during the differentiation and proliferation of human induced pluripotent stem cells (hiPSCs) can lead to tumorigenicity. This is a crucial concern in the development of stem cell-based therapies to ensure the safety and efficacy of the final product. Moreover, conventional genetic stability testing methods are limited by low sensitivity, which is an issue that remains unsolved. In this study, we assessed the genetic stability of hiPSCs and hiPSC-derived cardiomyocytes using various testing methods, including karyotyping, CytoScanHD chip analysis, whole-exome sequencing, and targeted sequencing. Two specific genetic mutations in *KMT2C* and *BCOR* were selected from the 17 gene variants identified by whole-exome and targeted sequencing methods, which were validated using droplet digital PCR. The applicability of this approach to stem cell-based therapeutic products was further demonstrated with associated validation according to the International Council for Harmonisation (ICH) guidelines, including specificity, precision, robustness, and limit of detection. Our droplet digital PCR results showed high sensitivity and accuracy for quantitatively detecting gene mutations, whereas conventional qPCR could not avoid false positives. In conclusion, droplet digital PCR is a highly sensitive and precise method for assessing the expression of mutations with tumorigenic potential for the development of stem cell-based therapeutics.

## 1. Introduction

Stem cell-based therapies show promise for regenerative and rare disease treatment. [1]. In particular, human induced pluripotent stem cells (hiPSCs) have garnered attention as starting materials for cell therapy owing to their pluripotency and capacity for large-scale manufacturing [2]. However, the complex and diverse processing steps involved, such as expansion and differentiation into target cells, often lead to the occurrence of various genetic mutations and pose challenges in terms of genetic instability [3]. From a regulatory perspective, it is essential to assess the genetic stability of not only the source cells but also the final differentiated target cells during the manufacturing process.

To ensure the genetic stability of stem cell-based products, regulatory guidelines from the United States Food and Drug Administration (FDA), the European Medicine Agency (EMA), the Korea Ministry of Food and Drug Safety (MFDS) and the International Council for Harmonisation of Technical Requirements for Pharmaceuticals for Human Use (ICH) strongly recommend the use of appropriate testing methods [4,5,6,7]. Currently, the most commonly used methods for assessing genetic stability include karyotyping, fluorescence in situ hybridization (FISH), and comparative genomic hybridization (CGH) arrays. However, these conventional testing methods have limitations, including difficulties in handling large-scale cell differentiation, extended processing times, and low resolution, which make it difficult to detect small structural changes or subtle abnormalities at the chromosomal level and to rely on data related to certain genes or regions of interest. In addition, it is becoming increasingly important to understand the mechanisms that control DNA integrity as DNA damage/repair processes [8] or chromatinolysis [9] occur, which could eventually lead to cancer. This renders the detection of new variations or genetic abnormalities challenging [10,11,12]. To address these limitations and achieve a more sophisticated assessment of genetic stability, the introduction of high-resolution and sensitive testing methods is essential. Recently, optical genome mapping (OGM) and next-generation sequencing (NGS) have been increasingly used to overcome the limitations of conventional methods [13]. OGM is suggested as a suitable alternative to detect the genomic structural variations of the examined patient [14]. OGM detects genomic structural variants and monitors tumor-related copy number changes, potentially serving as biomarkers [14]. NGS methods offer a higher resolution than conventional genetic stability assessment methods, enabling the efficient analysis of large genomes in a shorter timeframe than, for example, Sanger sequencing. Thus, NGS has been widely utilized in the fields of genomics research, diagnostics, and drug development [15,16,17,18,19].

In this study, genetic variations arising during the expansion of hiPSCs and their differentiation into cardiomyocytes (CMs) were systematically examined using established techniques, such as karyotyping and chromosomal microarray analysis (CMA), along with recent contemporary analytical methods, including whole-exome sequencing (WES) and targeted sequencing. After the identification of genetic mutations, the detected variants were validated and scrutinized for false positives by droplet digital PCR (ddPCR). This study aimed to evaluate genetic stability by cultivating and differentiating cells across three batches, considering potential genetic mutations during manufacturing and anticipating variations across passages [20].

The results of this study are expected to enhance the accuracy and reliability of safety evaluations by comparing traditional tests with new technologies. Additionally, this study aims to contribute to the improvement and standardization of genetic stability assessments in stem cell therapy, providing a solid foundation for its clinical application. Furthermore, the validation of ddPCR effectiveness and the proposal of applicable criteria for assessing the stability of stem cell therapies are expected to contribute to enhanced standardization and regulatory compliance in genetic stability assessments.

## 2. Results

### 2.1. Generation and Cardiac Gene Expression of hiPSC-Derived CMs (hiPSC-CMs)

To compare the genetic stability of hiPSCs and hiPSC-CMs, depending on the passage of hiPSC expansion during biomanufacturing, we differentiated hiPSCs into CMs at three different passages. Furthermore, to compare genetic stability across batches, we differentiated three batches of hiPSCs for each passage (early, intermediate, and late) (Figure 1A). The hiPSC-CMs were successfully generated from both the early-batch (EB) and intermediate-batch (IB) hiPSCs. The contractile beating of hiPSC-CMs with multiple contractile points was observed from 2 weeks after differentiation in both the EB and IB passage groups, followed by synchronized beating at 4 weeks. After 4 weeks of differentiation, the hiPSC-CMs gathered and detached from the bottom of the plate, gradually forming a reticulated cardiac sheet (Figure 1B). Conversely, in the late-batch (LB) hiPSCs, cells detached from the bottom of the plate before reaching 2 weeks of differentiation, preventing successful differentiation. Consequently, an analysis of the LB group was not possible in subsequent experiments. The successful differentiation of hiPSCs into CMs was confirmed by mRNA expression patterns using reverse transcription quantitative real-time PCR (RT-qPCR). First, the expression level of the undifferentiated/pluripotent gene *POU5F1* was significantly decreased in hiPSC-CMs after 2–4 weeks of differentiation compared to that in hiPSCs (Figure 1C). For the cardiogenic mesoderm marker *ISL1*, the expression level increased in hiPSC-CMs at 2 weeks but subsequently decreased at 4 weeks of differentiation (Figure 1D). The expression levels of myocardial sarcomeric protein-coding genes (*TNNT2*, *MYL2*, *MYH7*, *MYL7*, *MYH6*) significantly increased at 2 and 4 weeks of differentiation. Among these, *MYL2*, *MYH7*, and *MYH6* showed higher expression in hiPSC-CMs at 4 weeks than at 2 weeks (Figure 1E).

### 2.2. Cytogenetic Analysis

In the cytogenetic analysis of hiPSCs, G-banding was used to detect chromosomal variations and large structural abnormalities, which revealed no apparent cytogenetic abnormalities upon visual inspection (Figure 2A). These findings indicate a normal karyotype. However, CytoScanHD chip analysis, utilized for identifying subtle structural abnormalities, uncovered a 1.7 Mbps gain in genomic copy numbers at chromosome 20q11.21 (Figure 2B). The observed B-allele frequency changes at 20q11.21 suggested an unequal replication of the two alleles on chromosome 20. This finding was corroborated by alterations in the Log2Ratio values. Positive Log2Ratio values indicated an increase in DNA replication on the respective chromosomes. Thus, these changes in B-allele frequency and Log2Ratio indicate an increase in DNA quantity resulting from alterations in replication within the chromosomal region. The copy number variant (CNV) on chromosome 20q11.21 identified in hiPSCs encompassed the cancer-related gene *ASXL1*, as per the ClinVar database. Variants of this gene were detected across all groups (EB1, EB2, EB3, IB1, IB2, and IB3), regardless of cell passage and differentiation into CMs (Appendix A). This discovery suggests a pathogenic impact on the genetic or functional aspects of this region and highlights the persistence of this CNV in hiPSCs under different conditions.

### 2.3. Whole-Exome Sequencing

Using WES, we identified various single base-pair variants in the exons of hiPSCs and hiPSC-CMs. The average throughput depth coverage of the analyzed WES was approximately 100×. To identify putative high-impact mutations, we utilized ensemble variant effect predictor (VEP) software. Subsequently, upon verification with the Catalog of Somatic Mutations in Cancer (COSMIC) database, two mutations in the *MUC4* c.8032_8033insA; p. (Pro2678fs) and *KMT2C* c.2263C >T p. (Gln755*) genes were identified as tier 1 variants (Figure 3A, Table 1). Notably, these mutations were consistently detected in both early and intermediate passages of hiPSCs and persisted throughout the differentiation process for up to 4 weeks.

### 2.4. Targeted Sequencing

We detected genetic variants in the exons of 344 solid tumor-related genes in hiPSCs and hiPSC-CMs using targeted sequencing (Figure 3B). The average throughput depth of the targeted sequencing was 300×. A total of 15 variants were identified in 344 genes, including 9 missense, 3 nonsense, and 3 frame-shift mutations. Among them, *BRD4* c.3818G > A, *CEBPA* c566_568delinsACC, *KRT32* c.1205_1206 inv, *MADCAM1* c.800_801delinsCC, *MYC* c.857A > T, *PRIM2* c.857_860delinsCTTG, *RAD54L* c.1093_1169 + 15dup, *RREB1* c.2942T > G, and *TSC1* c.3106G > T consistently appeared, even when hiPSCs were passaged or differentiated. In contrast, the frequencies of genetic variants in *NOTCH4* c.17_18insTCTGCTG, *ZNF141* c.973_975delinsTCA, *ARID1A* c.1137 + 2T > A, *BCOR* c.1487_1500del, *FRG2B* c.31_33delinsTAG, and *SDHA* c.1415A > C varied depending on cell passage and differentiation. We particularly focused on the *BCOR* variant, which was absent in the early passages of hiPSCs but was detected in intermediate passages and persisted throughout the differentiation processes (Table 1). The identified variant caused a frame shift, which led to the premature termination of the BCL-6 corepressor protein. Pathogenic structural variations persisted in hiPSC-CMs, although they were not detected by karyotyping or CytoscanHD analysis.

### 2.5. Real-Time PCR and ddPCR

PCR analyses were conducted to quantify the precise expression levels of *KMT2C* and *BCOR* variants identified through WES and targeted sequencing. The *KMT2C* and *BCOR* variants in hiPSC-CMs were detected using each Taq-man MGB probe (Figure 4A). In contrast to the ddPCR results (Figure 4B), false-positive results were observed for both *KMT2C* and *BCOR* mutations in the no template control (NTC) and wild-type (WT) 10 fg groups using real-time PCR (Figure 4C), although they were undetectable. Despite employing identical primer/probe sets in real-time PCR and ddPCR, the expression levels of the *KMT2C* and *BCOR* variants in real-time PCR did not exhibit the mutant-type (MT) concentration-dependent increase in the WT 10 fg, WT 5 fg + MT 5 fg, and MT 10 fg groups. In the real-time PCR analysis, the detected levels of *KMT2C* variants were similar in all sample groups. In contrast, ddPCR analysis demonstrated an increase in *KMT2C* variant expression in the EB and IB groups, which was proportional to the differentiation period. The expression levels of the *BCOR* variants were significantly different between the EB and IB groups, according to both real-time PCR and ddPCR. In contrast to the targeted sequencing and ddPCR results, real-time PCR detected *BCOR* variants in the EB group, indicating false positives. *BCOR* variants were detected at higher levels in the IB group than in the EB group, regardless of PCR type. These results suggest that real-time PCR leads to false positives and unclear intergroup comparisons when applied to the detection of low-copy variants, whereas ddPCR facilitates a more accurate detection of small variants, such as *KMT2C* and *BCOR*.

### 2.6. Validation

#### 2.6.1. Precision

Precision was evaluated for the detection of genetic variants using four groups: NTC, WT 10 fg, WT 5 fg + MT 5 fg, and MT 10 fg. Inter-person precision involved two analysts (A and B) who independently measured samples three times each, with a total of six repetitions. The relative standard deviation (%RSD) was calculated as the mean of the variant copies and the standard deviation of the population. The %RSD values were 3.13–5.67% for *KMT2C* and 8.01–8.75% for *BCOR*, indicating precision between the analysts because the coefficient of variation (%RSD) value was’ within 20% (Table 2).

#### 2.6.2. Robustness

Robustness was assessed for the test method under eight different annealing temperatures for PCR, ranging from 53 to 57 °C. Three groups were measured: WT 10 fg, WT 5 fg + MT 5 fg, and MT 10 fg. The R^2^ value for the slope of copies/µL was used to assess robustness at each temperature. The results showed reliable data, regardless of annealing temperature. *KMT2C* and *BCOR* robustness testing indicated that the results were consistent across the annealing temperature range (53–57 °C), with all determination coefficients (R^2^ values) being 0.99 or higher.

#### 2.6.3. Limit of Detection (LOD)

To evaluate the LOD, we prepared 12 analytical samples by conducting a two-fold serial dilution of the MT control. The LOD estimation test was performed in triplicate at least three times. The %RSD values of the results from three repeated experiments were examined, and the lowest concentration with %RSD < 5% was determined to be the LOD. Based on this criterion, the LOD for the *KMT2C* variant was determined to be 6.78 copies/µL, while the LOD for *BCOR* was 3.82 copies/µL (Figure 5B).

#### 2.6.4. Specificity

We prepared five different test groups with the expected ratios by mixing the WT and MT controls: 1 to 0, 0.75 to 0.25, 0.5 to 0.5, 0.25 to 0.75, and 0 to 1. In each group, gene mutations were specifically expressed, depending on the ratio of the MT. Both the *KMT2C* and *BCOR* assays showed determination coefficients exceeding 0.99 in the linear regression line, confirming the ability of the assays to selectively detect genetic mutations (Figure 5C).

In summary, precision, robustness, LOD, and specificity testing demonstrated the reliable and consistent performance of the genetic analysis method, indicating its suitability for detecting genetic variants under different conditions and between different experimenters.

## 3. Discussion

The selection of starting materials for stem cell-derived therapy is a crucial step in the application of regenerative medicine. In particular, the number of passages of the starting cells is an important factor. Numerous studies have underscored the impact of the passage number of source cells on both the differentiation efficiency and functionality of the final differentiated product [21,22,23]. However, researchers have established passage numbers that vary widely based on the final differentiated cell type and donor information, and there are currently no standardized criteria. In this study, three passages of hiPSCs were used to differentiate the cells into CMs. Among them, late-passage hiPSCs failed to differentiate, as they detached from the plate surface before mesoderm formation. The failure to differentiate was attributed to cellular senescence. Based on this, subsequent genetic stability assessments were conducted in the early and intermediate passage groups.

The reprogramming of hiPSCs has been reported to lead to the deletion of tumor suppressor genes, posing a hurdle for the clinical application of cell therapies [24,25]. Moreover, it has been reported that significant and minor variations can occur during passaging and the differentiation process [26]. Therefore, we evaluated the genetic stability of identical hiPSC and hiPSC-CM samples using four different methods. Karyotyping results indicated the chromosomal stability of the hiPSCs. In contrast, CytoscanHD analysis (Affymetrix, Santa Clara, CA, USA) indicated a 1.7 Mbp large insertion variant in chromosome 20, specifically in the *ASXL1* gene, which was not observed in the karyotyping analysis. The *ASXL1* gene is involved in histone modification and chromatin remodeling, and heterozygous mutations in this gene can lead to premature truncations, potentially resulting in myeloid leukemia and Bohring–Opitz syndrome [27,28]. The utilization of the hiPSC line with this mutation as a source cell for regenerative medicine raises concerns about its potential for cancer induction.

Although the CytoscanHD chip method has the advantage of having a higher resolution than karyotyping, it has limitations in detecting small and rare single-nucleotide variations (<25 bp) that are not targeted by probes embedded in the chip [10]. To compensate for this, NGS methods have recently been used to determine the genetic stability of stem cell therapies [13]. In this study, we investigated small variations, such as structural variations and single-nucleotide variations, which were not identified by cytogenetic analysis, using targeted sequencing and WES methodologies. *BCOR* encodes an epigenetic regulator involved in cell differentiation that condenses chromatin during chromatin remodeling [29,30]. Mutations in the chromatin regulator *BCOR* have been reported as causative factors for the development of neural and hematological tumors, non-small cell lung cancer, and endometrial carcinoma [29,31,32]. Additionally, genetic variants of *KMT2C* identified through WES have been reported to interfere with the process of opening chromatin in the DNA repair system, potentially leading to tumor development [31,32]. Based on these findings, it is evident that even 1 bp genetic variants can pose a risk of tumorigenicity. Thus, high-resolution genetic screening is essential for utilizing the hiPSCs as starting cells for cell-based therapies.

Nevertheless, employing NGS as a method for genetic variation detection in clinical contexts has several limitations, including the sensitivity of the mutation detection [33], complexities in genomic region sequencing [34,35], limitations in databases for interpreting novel or rare mutations [36], restrictions in identifying structural genetic variations and copy number variations owing to coverage limitations [37], and the potential occurrence of false positives [38]. Thus, secondary validation using PCR is imperative for variants discovered by NGS.

In this study, the presence and expression levels of variants identified by WES and targeted sequencing were confirmed by real-time PCR and ddPCR. In contrast to the real-time PCR results, the ddPCR results showed no false positives in the control group. Moreover, in the hiPSC-CM group (at 0, 2, and 4 weeks), the ddPCR results clearly demonstrated a significant increase in variant expression levels during the differentiation period. These results were achieved through the execution of independent PCR within numerous droplets, ranging from thousands to tens of thousands. Furthermore, the automated procedures involved in droplet formation and result analysis mitigate experimental inaccuracies, thereby ensuring the consistency and reproducibility of the results [39,40]. Consequently, ddPCR has extensive applications in diverse domains, including environmental surveillance [41,42], pharmaceutical candidate substance evaluation [43], and quality and safety evaluation within the food industry [44]. However, previous studies have shown that ddPCR follows the principles of a Poisson distribution, which limits its upper quantification range compared to real-time PCR [45]. Positive droplets may become saturated at high template concentrations. To improve accuracy, the sample can be diluted for analysis or real-time PCR can be considered.

This study underscores the inadequacy of relying solely on NGS outcomes to detect minute genetic variations during hiPSC cultivation and CM differentiation. Instead, we advocate the imperative use of ddPCR. However, a significant challenge in assessing genetic stability is the absence of established criteria for ddPCR, which is a highly sensitive and precise method. To address this gap, we conducted a validation study to establish the reliability and applicability of ddPCR in scrutinizing genetic stability, particularly in hiPSCs for cell therapy. Our results demonstrated precision among experimenters, robustness under various annealing temperatures, and high reliability based on the LOD value. Moreover, specificity testing confirmed the ability of ddPCR to selectively detect specific genetic variations.

Although this study focused primarily on the expression of *BCOR* and *KMT2C*, it is necessary to substantiate the additional genetic variations identified through NGS analysis in future studies. Furthermore, elucidating the clinical implications of the identified mutations requires an analysis of their actual influence on mRNA and protein expression, along with their impact on metabolism. Hence, further studies should systematically explore the functional ramifications of the detected mutations and address the potential considerations arising from their therapeutic use. 

Furthermore, it is crucial to evaluate the pathogenicity of various mutations identified through NGS. To assess the correlation between pathogenicity and identified mutation, currently, several risk assessments are conducted by using the COSMIC database and various in silico tools such as PolyPhen-2 (Polymorphism Phenotyping v2), MutPred2, SIFT (Sorting Intolerant From Tolerant), and Mutation Taster. However, there is still a lack of established criteria for risk assessment, emphasizing the need for diverse assessment approaches and internationally harmonized standards. 

In summary, to evaluate cell therapy stability, it is important to employ diverse analytical methodologies in a complementary manner to screen the entire DNA region. Additionally, to validate these results, we propose the utilization of ddPCR as a precise and reliable methodology. In this study, the genetic variations detected between the two passages of hiPSC-CMs exhibited divergence. However, inter-batch variations in genetic variant disparities were not significantly different. Hence, we propose the continuous utilization of ddPCR for the quality control of differentiated cells following the screening of genetic mutations in the starting cells.

## 4. Materials and Methods

### 4.1. hiPSC Culture

Parental hiPSCs (FSiPS1, passage number #32, obtained from the National Stem Cell Bank, KNIH, Cheongju, Republic of Korea), which were reprogrammed from adult human fibroblasts, were cultured in Essential 8 (E8) medium (Gibco, Carlsbad, CA, USA) at 37 °C and 5% CO_2_. For xeno-free conditions, vitronectin recombinant human protein (Gibco) was used to coat the six-well plates. In the process of initially thawing cryopreserved hiPSC stock, the ROCK inhibitor (5 μM Y-27632) was used to enhance the recovery of cells. Subsequently, the ROCK inhibitor was removed, and passaging was conducted. The culture medium was changed daily, and the cells were subcultured (1:7 ratio) using a Gentile non-enzymatic cell dissociation method with Versene solution (Gibco) every 5 days. In the subsequent differentiation process, we divided hiPSCs into three groups based on passages: (ⅰ) EB (Passage 1) from the initial thawing of hiPSCs immediately after obtaining the parental hiPSCs (FSiPS1) from KNIH, (ⅱ) IB (Passage 11) from 10 passages after the initial thawing of FSiPS1, and (ⅲ) LB (Passage 21) from 20 passages after the initial thawing of FSiPS1. Three independent cell differentiation experiments were performed for each group.

### 4.2. Differentiation of hiPSCs into CMs

The hiPSC-CMs were generated by differentiating hiPSCs from three different passages into cardiac lineage cells using the Gibco^TM^ PSC Cardiomyocyte Differentiation Kit (Gibco), according to our previous report [46]. Briefly, hiPSCs were detached from the six-well plate using Accutase^TM^ (Innovative Cell Technology, San Diego, CA, USA) and resuspended for singularization. Then, 1 × 10^5^ cells/well were seeded in a vitronectin-coated 12-well plate in E8 medium. The E8 medium was replaced every day for 4 days to allow for the sufficient proliferation of hiPSCs, and then the following media were sequentially applied: CM differentiation mediums A and B and CM maintenance medium. After 10 days of differentiation, the beating of CMs was observed.

### 4.3. Karyotyping

hiPSCs were cultured in a vitronectin-coated T-25 flask, and KaryoMAX^TM^ Colcemid^TM^ Solution (Gibco) was added when the culture reached 80% confluence. After incubating at 37 °C and 5% CO_2_ for 1 h, the cells were detached with Accutase^TM^ (Innovative Cell Technology). After centrifugation, the cell pellets were gently resuspended in 5 mL of 0.075 M potassium chloride solution (Sigma, St. Louis, MO, USA) and incubated at 37 °C for 25 min. After cell fixation with Carnoy’s fixative solution (ratio of methanol to acetic acid = 3:1), the supernatant was removed by centrifugation. The cell pellet was placed on a slide and treated with 50% H_2_O_2_ (Sigma) at 24 °C for 3 min, followed by incubation at 60 °C for 30 min. After incubation, the slides were stained with Giemsa solution (Sigma). A karyotyping analysis of hiPSCs was performed using Gendix Software (Seoul, Republic of Korea).

### 4.4. CytoscanHD Chip Analysis

To detect CNVs, genomic DNA (gDNA) was extracted from hiPSC-CMs using a QIAamp DNA Mini Kit (QIAGEN, Hilden, Germany). Then, 250 ng of gDNA was digested with Nsp1 for 2 h at 37 °C. The digested DNA was purified and ligated with primer/adaptors at 16 °C for 3 h. Amplicons were generated by PCR using the primers provided by the manufacturer (Affymetrix, Santa Clara, CA, USA) on the ligation products. PCR was conducted according to the following protocol: 94 °C for 3 min, 30 cycles of 94 °C for 30 s, 60 °C for 45 s, and 65 °C for 15 s, followed by extension at 68 °C for 7 min. The PCR products were then purified and digested for 35 min at 37 °C to fragment the amplified DNA. The fragmented DNA was then labeled with biotinylated nucleotides through terminal deoxynucleotide transferase for 4 h at 37 °C. DNA was hybridized to a pre-equilibrated CytoscanHD chip (Affymetrix) at 50 °C for 16–18 h. After washing and scanning the CytoscanHD chips, data analysis was performed using AGCC software 4.0 (Affymetrix), followed by a filtration of cancer markers based on the pathogen region from ClinVar.

### 4.5. Whole-Exome Sequencing 

One microgram of input gDNA used in the targeted sequencing analysis was fragmented using an LE220 focused-ultrasonicator (Covaris, Woburn, MA, USA) to produce fragments of 150–200 bp. The fragmented gDNA samples were enriched using SureSelect^XT^ Human All Exon V6 (Agilent Technologies, Santa Clara, CA, USA), according to the manufacturer’s recommendations. At each step of library enrichment, gDNA was purified using AMPure XP beads (Beckman Coulter, Krefeld, Germany). The quantification and quality of the library were measured with the Quant-IT^TM^ PicoGreen^TM^ dsDNA reagent and kit (Thermo Fisher Scientific, Waltham, MA, USA) and 1% agarose gel electrophoresis. Each DNA library was hybridized with SureSelect^XT^ Human All Exon Capture Bait (Agilent Technologies) and eluted using Dynabeads^TM^ MyOne^TM^ Streptavidin T1 (Invitrogen, Waltham, MA, USA). The captured library was amplified by Veriti^TM^ 96 well Thermal Cycler (Applied Biosystems, Waltham, MA, USA) and qualified using the TapeStation DNA Screentape D1000 (Agilent Technologies). The amplified products were pooled in equimolar amounts and diluted to a final loading concentration of 10–15 nM according to the Sureselect^XT^ target enrichment system protocol. The final libraries were sequenced on a NovaSeq (Illumina, San Diego, CA, USA) platform, and the paired-end sequence data were mapped to the human reference genome using the BWA mapping program.

### 4.6. Targeted Sequencing

Total gDNA was extracted from hiPSCs and hiPSC-CMs at 0, 2, and 4 weeks after differentiation using a QIAamp DNA Mini Kit (QIAGEN). After the quantification of the gDNA concentration, 200 ng of each gDNA sample was digested using a SureSelect Enzymatic Fragmentation Kit (Agilent Technologies) to create a library of gDNA restriction fragments. The enzymatically fragmented gDNA samples were enriched using ONCO AccuPanel (NGeneBio, Seoul, Republic of Korea) according to the manufacturer’s recommendations. At each step of library enrichment, the gDNA was purified using AMPure XP beads (Beckman Coulter) to selectively bind nucleic acids based on their size. Prior to sample pooling, the quantity and quality of the library were measured using a Quibit dsDNA BR Assay Kit (Invitrogen) and 1% agarose gel electrophoresis. DNA libraries were pooled, hybridized with biotin-labeled RNA probes, and eluted using Dynabeads^TM^ MyOne^TM^ Streptavidin T1 (Invitrogen). The captured library was amplified by Veriti^TM^ 96 well Thermal Cycler (Applied Biosystems), and diluted to a final loading concentration of 1.5 pM. Final libraries with 1% PhiX control were sequenced at a paired-end 150 bp (2 × 150 bp) read length on the MiniSeq platform (Illumina, San Diego, CA, USA). The FASTQ file containing the sequence data was analyzed using NGeneAnalySys v1.6.4 software (NGeneBio).

### 4.7. Control Template Generation

Control DNA sequences containing the target loci were generated by cloning them into a pBHA vector (Bioneer, Daejeon, Republic of Korea). DH5α competent cells (Thermo Fisher Scientific) were transformed with the resulting plasmids using a heat-shock procedure at 42 °C for 90 s, cultured on a selective agar plate with ampicillin (50 µg/mL), and incubated at 37 °C overnight. Plasmid DNA (pDNA) was extracted using an Exprep Plasmid SV Mini Kit (GeneAll, Seoul, Republic of Korea). The isolated pDNA was digested with the Bsal-HFv2 restriction enzyme (NEB, Ipswich, MA, USA) and purified using the Expin CleanUp SV Mini Kit (GeneAll). The final DNA fragments were verified by 1% agarose gel electrophoresis to confirm the successful generation of the control template DNA.

### 4.8. ddPCR Analysis

Total gDNA was extracted from hiPSCs and hiPSC-CMs at 0, 2, and 4 weeks after differentiation using a QIAamp DNA Mini Kit (QIAGEN), according to the manufacturer’s recommendations. After the quantification of the gDNA concentration, the ddPCR assay was conducted in a 20 μL reaction volume, comprising 10 μL ddPCR supermix for probes (no dUTP; Bio-Rad Laboratories, Hercules, CA, USA), 1 μL of the template DNA sample, 900 nM of each primer (forward/reverse; Thermo Fisher Scientific), and 250 nM of each MGB probe (VIC/FAM; Thermo Fisher Scientific). The primer and probe sequences for the target genes are shown in Appendix A. The ddPCR mixture was loaded on ddPCR 96-well semi-skirted plates (Bio-Rad Laboratories). The plate was placed in an Automated Droplet Generator (Bio-Rad Laboratories) to partition the sample into droplets using Automated Droplet Generation Oil for Probes (Bio-Rad Laboratories). Next, the PCR was run using a thermal cycler (T100, Bio-Rad Laboratories) with the following cycling conditions: 95 °C for 10 min, 40 cycles of 94 °C for 30 s and 55 °C for 1 min, and 98 °C for 10 min. The number of droplets with or without a mutant target was read by a Droplet Reader (QX200, Bio-Rad Laboratories). The absolute number of copies was calculated using QX Manager Software 2.1 Standard Edition (Bio-Rad Laboratories) according to the Poisson correction. The quantification measurements of the target variants were presented as the copies/µL of the sample.

### 4.9. Evaluation of Differentiation into Cardiomyocyte Using RT-qPCR

Total RNA from hiPSCs and hiPSC-CMs at 0, 2, and 4 weeks after differentiation was extracted using the RNeasy Plus Mini Kit (QIAGEN) according to the manufacturer’s recommendations. After dissolving the final product in DEPC-treated water, the RNA concentration was quantified. Complementary DNA (cDNA) was synthesized using 1 μg RNA and an iScript™ cDNA Synthesis Kit (Bio-Rad Laboratories) according to the manufacturer’s instructions. RT-qPCR was performed using a cDNA template and a QuantiTect SYBR Green PCR Kit (QIAGEN). The PCR was run using a real-time PCR system (7900HT Fast Real-Time PCR System, Applied Biosystems) with the following cycling conditions: 50 °C for 2 min and 95 °C for 15 min, followed by 40 cycles of 94 °C for 15 s, 60 °C for 30 s, and 72 °C for 30 s. The primer sequences for the target genes are listed in Appendix A. The relative expression of genes was calculated and expressed as 2^−ΔΔCt^ using ExpressionSuite Software Version 1.2 (Thermo Fisher Scientific). Expression values were normalized to the expression of 18S rRNA as a housekeeping gene.

### 4.10. Detection of Tumorigenic Variants Using Real-Time PCR

The quantification of variant copies obtained from ddPCR was compared with that of custom MGB primers/probes for real-time PCR using the same amount of gDNA. The 25 μL real-time PCR mixture consisted of 12.5 μL TaqPath ProAmp master mix (Thermo Fisher Scientific), 1 μL of DNA template, 900 nM of each primer (forward/reverse; Thermo Fisher Scientific), and 200 nM of each MGB probe (VIC/FAM; Thermo Fisher Scientific). The PCR was run using a real-time PCR system (7900HT Fast Real-Time PCR System, Applied Biosystems) with the following cycling conditions: 60 °C for 30 s for pre-read, 95 °C for 5 min, 40 cycles of 94 °C for 15 s, and 60 °C for 1 min for amplification, followed by 60 °C for 30 s for post-read. The quantification of the target variants was performed using SDS 2.4 software (Thermo Fisher Scientific).

### 4.11. Validation of ddPCR for hiPSC-CMs

In accordance with the ‘Validation of Analytical Procedures Q2’ of the ICH guidelines, the validation parameters necessary for the detection of genetic variants using ddPCR were determined and validated. ddPCR was determined to be suitable for the quantitative analysis of each genetic variant. The following parameters were validated: specificity, precision, robustness, and LOD. The primer and probe sequences for the target genes are shown in Appendix A.

## 5. Conclusions

Our study highlights the crucial role of high-resolution genetic analysis, including WES and targeted sequencing, followed by the validation of identified genetic mutations with ddPCR in stem cell-based cell therapy. During the differentiation of hiPSCs into cardiomyocytes, genetic variants have been observed through various genetic analyses. The identification of potential tumorigenic mutations underscores the need for robust genetic safety evaluations in hiPSC-based cell products. Validation through ddPCR not only confirms these mutations but also establishes a reliable method for assessing the genetic stability of hiPSC-derived cardiomyocytes. We propose the integration of high-resolution assays into standard safety evaluation protocols for hiPSC-based therapies, emphasizing the use of orthogonal validation methods to verify the potential tumorigenicity associated with genetic variants. This comprehensive approach is pivotal for advancing the translational potential of hiPSC-based cell therapeutics, ensuring safety and supporting their potential clinical applications. 

## Figures and Tables

**Figure 1 ijms-25-01101-f001:**
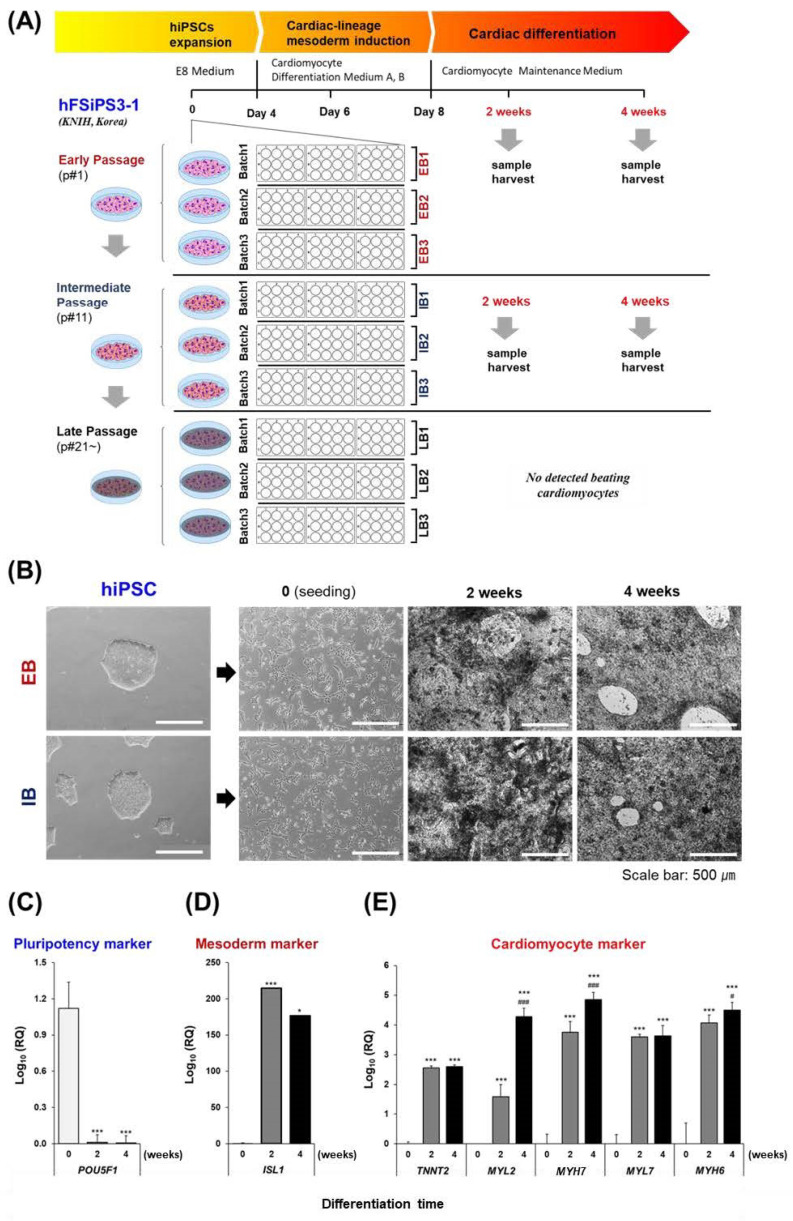
Experimental scheme and assessment of hiPSC-CM differentiation. (**A**) Experimental Design: hiPSCs from three passages (EB, IB, or LB) were subjected to CM differentiation for 0, 2, or 4 weeks. (**B**) Time-lapse images of CMs at different stages of differentiation were obtained using an inverted microscope at 4× magnification, and the scale bar represents 500 µm. (**C**) Quantification of *POU5F1* expression levels in undifferentiated hiPSCs. (**D**) Quantification of *ISL1* expression levels, a marker associated with mesoderm, during hiPSC differentiation. (**E**) Quantification of gene expression levels associated with cardiomyocytes (*TNNT2*, *MYL2*, *MYH7*, *MYL7*, *MYH6*) during the differentiation process. * indicates *p* < 0.05, *** indicates *p* < 0.001 compared to the 0 week of each hiPSC-CM group. ^#^ indicates *p* < 0.05, ^###^ indicates *p* < 0.01 compared to the 2 weeks of each hiPSC-CM group. hiPSCs, human-induced pluripotent stem cells; CMs, cardiomyocytes; hiPSC-CMs, hiPSC-derived CMs; EB, early batch; IB, intermediate batch; LB, late batch; *POU5F1*, POU class 5 homeobox 1; *ISL1*, ISL LIM homeobox 1; *TNNT2*, troponin T2; *MYL2*, myosin light chain 2; *MYH7*, myosin heavy chain 7; *MYL7*, myosin light chain 7; *MYH6*, myosin heavy chain 6; RQ, relative quantification.

**Figure 2 ijms-25-01101-f002:**
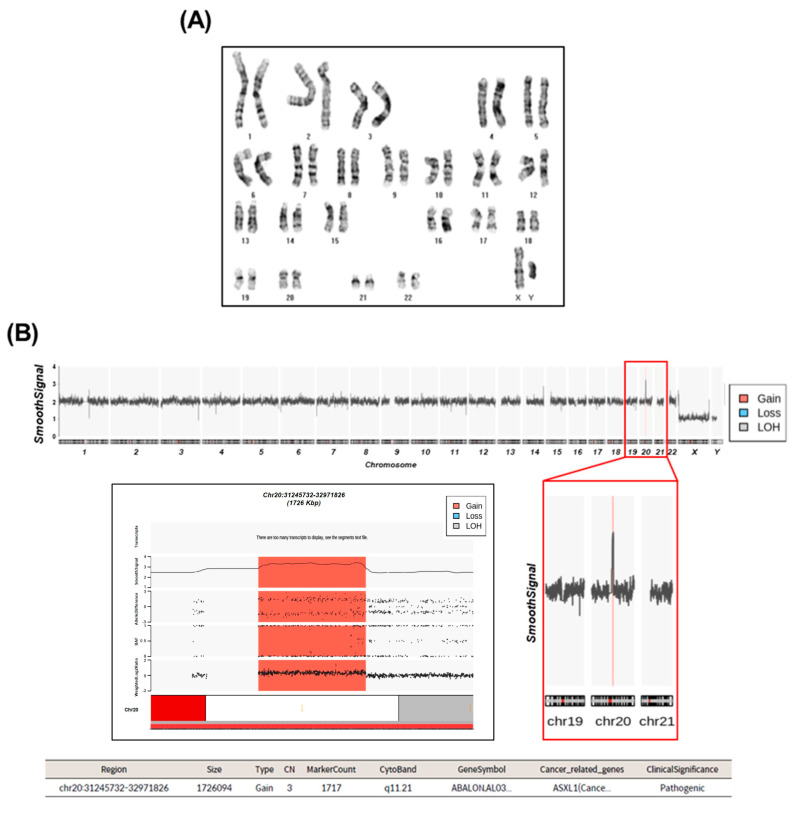
Cytogenetic analysis of hiPSCs. (**A**) G-banding karyotyping results display the overall structure of chromosomes. (**B**) CytoscanHD results illustrate chromosome structure and changes in copy numbers. In the smooth signal of the entire chromosome, the orange line indicates a ‘Gain’. The magnified view of this region (highlighted by the red box) precisely reveals the location of the variant within chromosome 20. B-allele frequency, weighted Log2Ratio, and allele difference data elucidate intricate variations in copy numbers in the identified mutation region. The table provides information on genetic details and clinical significance associated with the copy number variant using ClinVar. LOH, loss of heterozygosity; CN, copy number.

**Figure 3 ijms-25-01101-f003:**
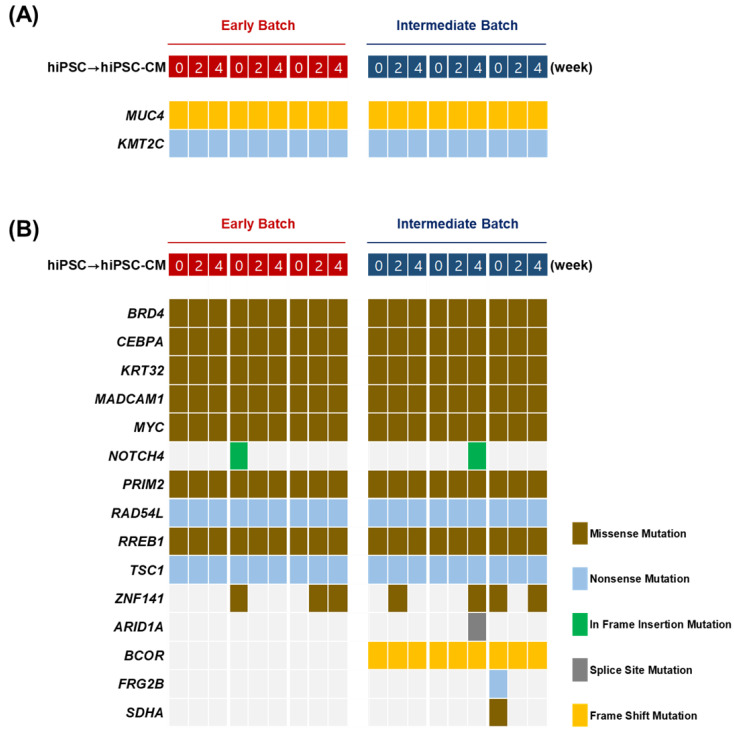
Next-generation sequencing in hiPSCs and hiPSC-CMs. (**A**) Mutations identified by WES. (**B**) Mutations identified by targeted sequencing. Genomic alterations are annotated according to the color panel on the right of the image. Missense mutation (brown), nonsense mutation (sky blue), in-frame insertion mutation (green), splice site mutation (gray), and frame-shift mutation (yellow). The time frame for the analysis of samples is shown at the top of the image from early batch (red) and intermediate batch (blue), and the gene names of each identified mutation are listed on the left side of the image.

**Figure 4 ijms-25-01101-f004:**
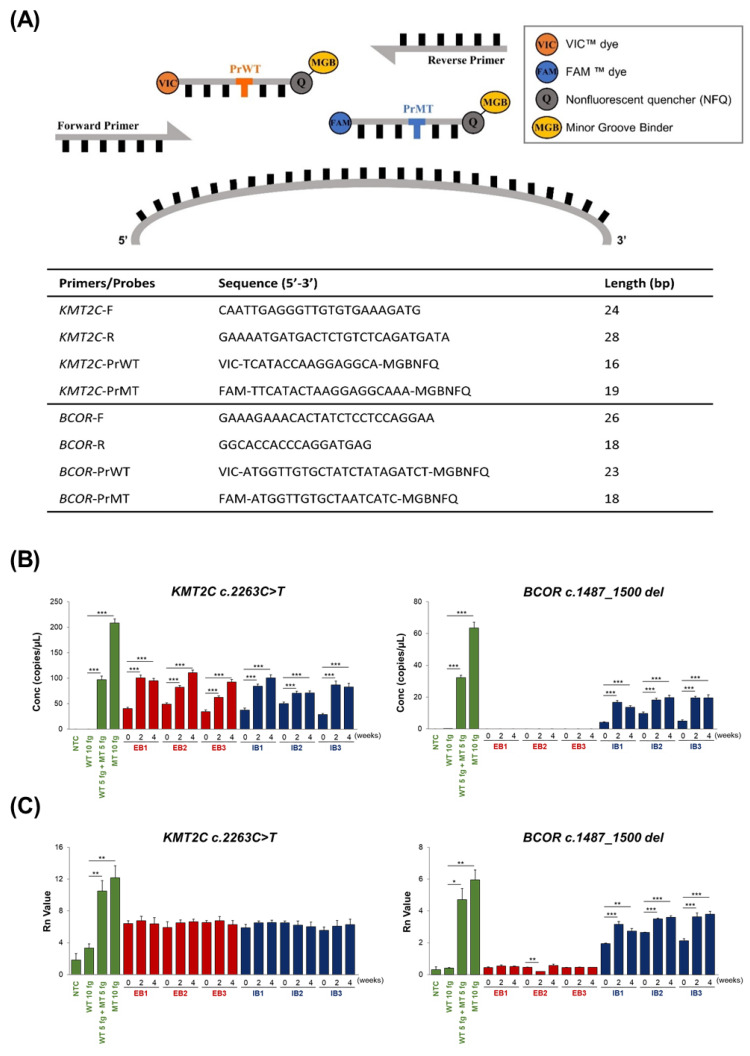
*KMT2C* and *BCOR* variants expression using MGB TaqMan probes. (**A**) Illustration and sequences of primers and TaqMan MGB probes designed to determine the expression rates of the *KMT2C* and *BCOR* genes in wild-type (WT) and mutant-type (MT) conditions. The primer sequences are identical for both WT and MT. For the probes, WT is labeled with the VIC fluorescent dye (orange circle), while MT is labeled with the FAM fluorescent dye (blue circle). To enhance specificity, fluorescently labeled probes are quenched using the MGB (yellow circle)-eclipse quencher (gray circle). These probes are denoted PrWT and PrMT. Detailed sequences are provided in the table below. The variants were measured by two different molecular techniques: (**B**) ddPCR and (**C**) real-time PCR. The data represent the means ± SD of triplicates; * indicates *p* < 0.05, ** indicates *p* < 0.01, *** indicates *p* < 0.001 compared to the WT 10 fg group or 0 week of each hiPSC-CM group. VIC, VIC fluorophore; FAM, FAM fluorophore; Q, nonfluorescent quencher; MGB, minor groove binding; F, forward; R, reverse; PrWT, probe for wild type; PrMT, probe for mutant type; ddPCR, digital droplet PCR; *KMT2C*, histone-lysine N-methyltransferase 2C; *BCOR*, B-cell lymphoma 6 protein corepressor; NTC, no template control; Conc, concentration; Rn, normalized reporter.

**Figure 5 ijms-25-01101-f005:**
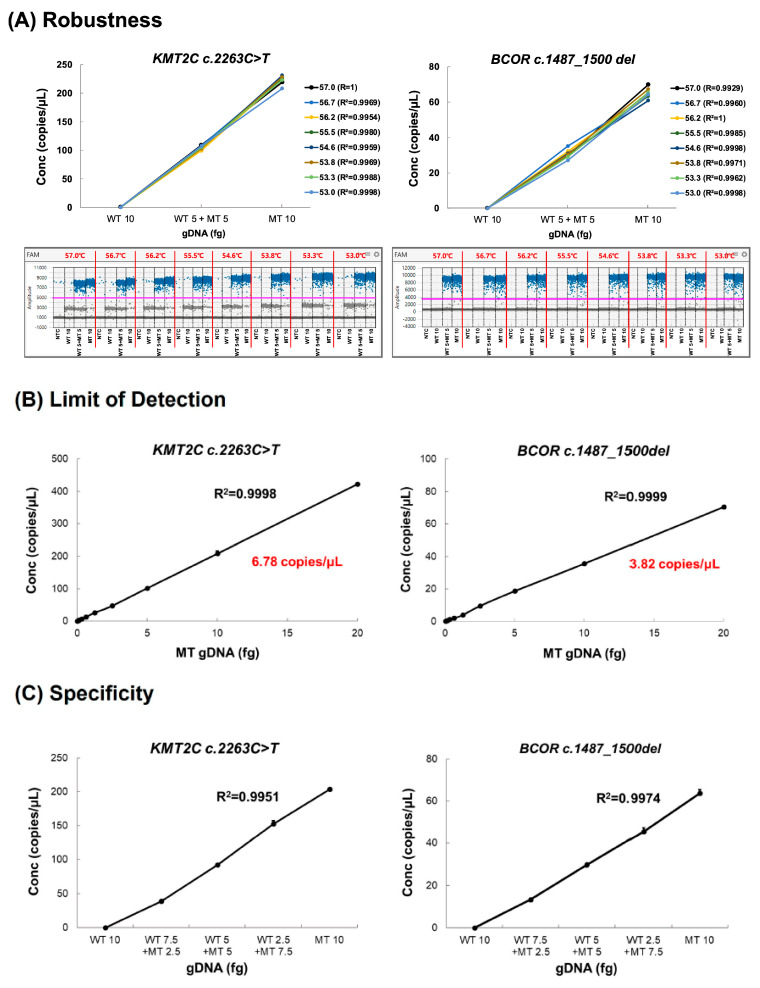
Validation of *KMT2C* and *BCOR* variants using ddPCR. (**A**) Robustness. The control sample was run by ddPCR using several temperatures for the annealing/extension steps (53–57 °C). The positive droplets are represented in blue dot, whereas negative droplets are in black dot. Red lines indicate the separation between the different temperatures, and the pink line represents the set fluorescence threshold to distinguish positive and negative droplets. (**B**) LOD. Two-fold dilutions of the MT control were run in three replicates. The LOD was defined as the lowest concentration of target genetic variation that can be detected with a %RSD <5%. (**C**) Specificity. The R^2^ results from the experiments with gradient mixed control were >0.99. Data are presented as means ± SD. LOD, limit of detection; Conc, concentration.

**Table 1 ijms-25-01101-t001:** Detection of *KMT2C* and *BCOR* variants using WES and targeted sequencing.

*GENE*	Consequence	Protein Change	Variant Type	Tier (COSMIC)	Genomic Change	Chrom
*KMT2C*	Nonsense_Mutation	p.Gln755Ter	SNV	-	c.2263C > T	chr7
*BCOR*	Frame_Shift_Del	p.Ile496AsnfsTer17	Del	T3	c.1487_1500del	chrX

**Table 2 ijms-25-01101-t002:** Inter-person precision of *KMT2C* and *BCOR* variants using ddPCR.

Sample Type	*KMT2C* c.2263C > T Concentration (Copies/µL)							
Analyst	A-1	A-2	A-3	B-1	B-2	B-3	Mean ± SD	%RSD
NTC	0.00	0.00	0.00	0.00	0.00	0.00	0.00 ± 0.00	-
WT 10 fg	0.00	0.00	0.00	0.00	0.00	0.00	0.00 ± 0.00	-
WT 5 fg + MT 5 fg	93.92	91.73	94.38	103.07	99.30	107.51	98.32 ± 5.57	5.67
MT 10fg	210.82	205.19	199.40	212.75	217.00	218.40	210.59 ± 6.60	3.13
**Sample Type**	** *BCOR* ** **c.1487_1500del Concentration (copies/µL)**							
Analyst	A-1	A-2	A-3	B-1	B-2	B-3	Mean ± SD	%RSD
NTC	0.00	0.00	0.00	0.00	0.00	0.00	0.00 ± 0.00	-
WT 10 fg	0.00	0.00	0.00	0.00	0.00	0.00	0.00 ± 0.00	-
WT 5 fg + MT 5 fg	25.61	29.26	30.03	31.57	33.30	31.50	30.21 ± 2.42	8.01
MT 10fg	55.82	61.72	53.46	64.83	69.50	62.20	61.25 ± 5.36	8.75

## Data Availability

The data supporting the findings of this study are available from the corresponding author upon request.

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
