# Peer review of "Assessment of Genetic Stability in Human Induced Pluripotent Stem Cell-Derived Cardiomyocytes by Using Droplet Digital PCR"

_ijms, 2024, doi:10.3390/ijms25021101_

Round 1

Reviewer 1 Report

Comments and Suggestions for Authors

Generation of iPSC from adult host tissue cells has brough great hopes for improved therapeutic outcomes in tissue regenerative efforts, however the method has been met with equally great skepticism regarding the safety of iPSC application in gene therapy due to accumulation of genetic mutations in the process of iPSC propagation and differentiation into target cells. For this reason, studies on genetic stability of iPSC and iPSC-derived tissue cells are of great importance.

The work presented in this manuscript has been methodically designed to evaluate the accuracy and sensitivity of several well-established methods for genome analysis, including karyotyping, chromosomal microarray analysis (CMA), whole-exome sequencing (WES), and targeted sequencing of the ONCO gene panel. Quantitative (qPCR) and droplet digital PCR (ddPCR) were applied for validation of select gene copy numbers in early and late passage number populations of iPSC compared to the paternal cell populations.

This study is important to the field of therapeutic safety evaluation with its demonstration that karyotyping is not a sensitive method for investigation of genomic stability and needs to be substituted with methods incorporating next generation genome sequencing analysis which are capable of pinpointing genomic alternations at a gene locus level. With WES and targeted sequencing of the ONCO gene panel, 17 genes were identified to have accumulated mutations of various kind thus confirming the suspected possibilities of increased risk of cancer development from iPSC with high passage numbers.

While it is important to demonstrate accumulation of mutations in genes regulating chromosomal arrangement and genomic stability (BCOR and KMT2C), it would be more impactful for this study to prove that these mutations have actual effect on gene function. Such proof is missing from this study, instead, a waiver has been provided in the Discussion section (lanes 323-326).

Once the importance of certain genomic mutations in the performance of gene function is demonstrated (which is not the focus of this study) high sensitivity method for detection of genomic variants would have sensible power in the clinical diagnosis.

 In this study, ddPCR has been promoted as a highly sensitive method for gene variant copy number calculation that outperforms qPCR. Adequate controls have been included to prove the advantage of the method in detection of low-copy number of gene variants amongst various populations of iPSC and their CM derivatives.

Since great emphasis has been made on the ddPCR as a validation method in iPSC genomic stability, the authors have included graphical explanation on the design of primers and probes capable of differentiation between gene variants in the Supplementary material. It would be beneficial to include a graphic in Fig 4 of the target genes with indication of the accumulated mutations and the positioning of the primers and probes used in the qPCR and ddPCR.

Needs fixing or additional information:

1). In section 2.5, the authors indicate that RT-qPCR was used to evaluate gene copy number, however in the Methods (4.9.2) it is explained that gDNA has been applied (not RNA) so there is no RT step. The only RT-qPCR in this study is done when evaluation of CM differentiation was done.

2). It is important to include in the Methods section of this manuscript information on the passage number of the parental iPSC cultures obtained from the repository (KNIH) and whether ROCK inhibitor was added to the iPSC media during cell passaging.

3). In section 4.3 describing the method for karyotyping in lane 367 there is a statement that Accutase was used to digest the cells, when the enzyme only detaches the cells.

Reviewer 2 Report

Comments and Suggestions for Authors

The manuscript titled “Assessment of Genetic Stability in Human-induced Pluripotent Stem Cell-derived Cardiomyocytes by Using Droplet Digital-PCR” by Park, J.W.; et al. is a scientific work where the authors study the positive impact in terms of data acquisition accuracy in the use of droplet digital PCR to monitor the genetic variations of pluripotent stem cells and their differentiation to cardiomyocytes. The authors carried out a validation procedure of two specific genetic mutations (BCOR and KMT2C) according to the International Council Harmonisation (ICH) standards.

However, it exists some points that need to be addressed (please, see them below detailed point-by-point). The most relevant outcomes remarked by the authors can contribute in the growth of many fields like the design of the next-generation of cancer therapies exploiting genetic biomolecular targets. For this reason, I will recommend the present scientific manuscript for further publication in the International Journal of Molecular Sciences once all the below described suggestions will be properly fixed.

Here, there exists some points that must be covered in order to improve the scientific quality of the manuscript paper:

1) ABSTRACT. “The applicability (…) ICH guidelines (…)” (lines 19-21). Please, the abbreviation “ICH” should be defined by adding the full-name term “The International Council Harmonisation (for pharmaceuticals for human use)”. Same comment for all the policy-makers indicated in the line 40: “(…) from the FDA, EMA, MFDS, (…)”.

2) KEYWORDS. (OPTIONAL) The authors should consider to add the term “mutation detection sensitivity” in the keyword list.

3) INTRODUCTION. “Stem cell-based therapy is a promising therapeutic approach (…)” (line 30). Please, the authors should rephrase this sentence in order to avoid redundacies. This comment should be taken into account for the rest of the main manuscript body text.

4) “Currently, the most commonly used methods (…) fluorescence in situ hybridization (FISH), and comparative genomic hybridization (CGH) arrays” (lines 41-43). Here, it should be also discussed the optical genome mapping (OGM) as suitable alternative to detect the genomic structural variations of the examined patient [1]. This technology has already employed to monitor the copy number alterations which are linked to the tumor disease progress and could act as potential biomarkers.

[1] Yuan, Y.; et al. Advances in optical mapping for genomic research. Comput. Struct. Biotechnol. J. 2020, 18, 2051-2062. https://doi.org/10.1016/j.csbj.2020.07.018.

5) “However, these traditional (…) detect small structural changes or subtle abnormalities at the chromosomal level (…) regions of interest” (lines 43-47). Even if I agree with this statement provided by the authors, it is neccesary to strenghten the significance of this research field and the promising opened future avenues to gather the knowledge about the underlying mechanisms that control the DNA integrity as DNA damage/repair processes [2] or chromatinolysis [3] where could eventually induce cancer diseases.

[2] Lee, T-H.; et al. DNA Oxidation and Excision Repair Pathways. Int. J. Mol. Sci. 2019, 20, 6092. https://doi.org/10.3390/ijms20236092.

[3] Novo, N.; et al. Beyond a platform protein for the degradosome assembly: The Apoptosis-Inducing Factor as an efficient nuclease involved in chromatinolysis. PNAS Nexus 2022, 2, pgac312. https://doi.org/10.1093/pnasnexus/pgac312.

6) RESULTS. Figure 5 (line 234). The standard deviations (SD) bars should be added for each measured condition to determine the robustness, limit of detection and specificity of the KMT2C and BCOR variants.

7) Then, did the authors visualize any constraint in the data obtantion and interpretation according to the validation analysis (e.g. primer design challenges dealing with closely related genome sequences)? (lines 198-247). A brief statement should be added in this regard.

8) DISCUSSION. This section clearly states the most relevant outcomes found in this research. No actions are requested from the authors.

9) MATERIALS & METHODS. “4.1. hiPSC culture” (lines 344-354). Please, the authors should specify if the employed biology material comes from perinatal, embryonic or adult stem cells.

10) “4.9. RT-qPCR” (lines 452-477). Did the authors add some RNAse inhibitor to preserve the RNA integrity? A brief statement should be added in this regard.

11) CONCLUSIONS. “We suggest that an orthogonal validation method should be used to verify the possibility of tumorigenicity” (lines 491-492). Please, the authors should highlight some action lines to reach this goal. Finally, the references are in the proper format of the International Journal of Molecular Sciences (No actions are requested from the authors).

Comments on the Quality of English Language

The manuscript is generally well-written. However, it may be advisable to take a final check in order to polish those final details susceptible to be improved.
